# Characterization of Interactions between CTX-M-15 and Clavulanic Acid, Desfuroylceftiofur, Ceftiofur, Ampicillin, and Nitrocefin

**DOI:** 10.3390/ijms23095229

**Published:** 2022-05-07

**Authors:** Parvaneh Ahmadvand, Johannetsy J. Avillan, Jacob A. Lewis, Douglas R. Call, ChulHee Kang

**Affiliations:** 1Department of Chemistry, Washington State University, Pullman, WA 99164, USA; parvaneh.ahmadvand@wsu.edu (P.A.); jacob.lewis2@wsu.edu (J.A.L.); 2Paul G Allen School for Global Health, Washington State University, Pullman, WA 99164, USA; johannetsy.avillan@wsu.edu (J.J.A.); drcall@wsu.edu (D.R.C.)

**Keywords:** antimicrobial resistance, β-lactamase, CTX-M-15, inhibitor, antibiotic, β-lactam compounds, crystal structure, inhibition mechanism

## Abstract

Cefotaximase-Munich (CTX-M) extended-spectrum beta-lactamases (ESBLs) are commonly associated with Gram-negative, hospital-acquired infections worldwide. Several beta-lactamase inhibitors, such as clavulanate, are used to inhibit the activity of these enzymes. To understand the mechanism of CTX-M-15 activity, we have determined the crystal structures of CTX-M-15 in complex with two specific classes of beta-lactam compounds, desfuroylceftiofur (DFC) and ampicillin, and an inhibitor, clavulanic acid. The crystal structures revealed that Ser70 and five other residues (Lys73, Tyr105, Glu166, Ser130, and Ser237) participate in catalysis and binding of those compounds. Based on analysis of steady-state kinetics, thermodynamic data, and molecular docking to both wild-type and S70A mutant structures, we determined that CTX-M-15 has a similar affinity for all beta-lactam compounds (ceftiofur, nitrocefin, DFC, and ampicillin), but with lower affinity for clavulanic acid. A catalytic mechanism for tested β-lactams and two-step inhibition mechanism of clavulanic acid were proposed. CTX-M-15 showed a higher activity toward DFC and nitrocefin, but significantly lower activity toward ampicillin and ceftiofur. The interaction between CTX-M-15 and both ampicillin and ceftiofur displayed a higher entropic but lower enthalpic effect, compared with DFC and nitrocefin. DFC, a metabolite of ceftiofur, displayed lower entropy and higher enthalpy than ceftiofur. This finding suggests that compounds containing amine moiety (e.g., ampicillin) and the furfural moiety (e.g., ceftiofur) could hinder the hydrolytic activity of CTX-M-15.

## 1. Introduction

Multi-drug resistance among Gram-negative bacteria is a serious concern for public health worldwide [1]. The extended-spectrum β-lactamase (ESBL) CTX-M, which is commonly found with Enterobacteriaceae (*bla*_CTX-M_) [2,3], was originally reported as a plasmid-associated gene from *E. coli* collected from dogs in Japan (1986), and later reports documented its presence in human patients from Germany (1989), Canada (2000), and the United States (2001) [4,5,6,7]. CTX-M together with CMY are now the most prevalent Extended-spectrum cephalosporin-resistant enterobacteriaceae (ESCRE) found in *E. coli*, *K. pneumoniae*, and *Proteus mirabilis* [8].

β-lactam antibiotics are the most commonly used antibiotics [9]. These compounds are covalent inhibitors that prevent bacterial cell wall biosynthesis by inactivating the transpeptidase enzymes called penicillin-binding proteins (PBPs) [10,11,12]. PBPs are essential for the formation of the peptidoglycan layer of Gram-negative bacteria by catalyzing the transglycosylation and transpeptidation reactions. The active site for the transpeptidation reaction in PBPs is the drug target site of β-lactams. By mimicking the D-alanyl-D-alanine dipeptide, β-lactams establish a covalent bond with PBPs and generate an acyl-enzyme complex that inhibits the final transpeptidation reaction [13]. As a result, peptidoglycan polymers can no longer be cross-linked, resulting in cell wall lysis and cell death [14]. The structural integrity of the cephem/penam and β-lactam rings are key factors of antimicrobial activity for these compounds. Bacterial resistance to these compounds is primarily conveyed by production of β-lactamases (EC 3.5.2.6) that hydrolyze the amide bond of the 4-member ring in β-lactams and eliminates their antimicrobial activity [15,16,17,18,19,20].

Bacterial resistance to β-lactam compounds can be classified into four general categories: (1) alteration of PBP genes; (2) reduced drug entry into the periplasm, often achieved by downregulation of porins; (3) active efflux of imported antibiotics; and (4) inactivation of antibiotics by β-lactamase [21,22]. Among these, β-lactamases are the most important contributor to antimicrobial resistance because enzymes can deactivate β-lactam molecules faster than they are internalized. Consequently, the β-lactamases from ESKAPE pathogens (*Enterococcus faecium*, *Staphylococcus aureus*, *Klebsiella pneumonia*, *Acinetobacter baumannii*, *Pseudomonas aeruginosa*, and *Enterobacter* species) are the most important factor for bacterial resistance to β-lactam compounds [23,24,25].

Based on the individual sequence motifs and basic differences in enzymatic mechanisms, β-lactamases are classified into four classes (A, B, C, and D) [26]. Class A, C, and D are active serine proteases that are differentiated by a Ser-Xaa-Xaa-Lys motif that hydrolyzes the amide bond in the four-membered ring of β-lactam compounds to form acyl-enzyme complexes. Class B β-lactamases are zinc metalloenzymes that contain one or two zinc ions in the catalytic site. These enzymes use a His-Xaa-His-Xaa-Asp motif to cleave the amide bond in the four-membered ring of β-lactams, via a nucleophilic water that initiates the hydrolytic reaction [27]. All classes of β-lactamases are encoded on chromosomes or plasmids of Gram-negative bacteria including the ESKAPE pathogens [28]. Class A β-lactamases, including CTX-M, display a wide range of resistance to antibiotics such as penicillins, early cephalosporins, monobactams, carbenicillin, cefepime, and carbapenems. Class B has notable activity against cephalosporins, carbapenems, and penicillins, although they are inhibited by monobactams [29]. Class C β-lactamases display effective hydrolysis only for cephalosporins [30,31]. While class D includes β-lactamases that are active against cephalosporins and carbapenems.

This study investigated the interaction between CTX-M-15 and four bicyclic β-lactam compounds, which include cephalosporin and penicillin families, and one β-lactamase inhibitor. They share basic structural features including a five- or six-membered heterocyclic ring attached to a four-membered β-lactam ring that contains a secondary amino group and other substituents (Figure 1). Ceftiofur is a veterinary, third-generation cephalosporin β-lactam that is used to treat respiratory infections, metritis, and foot rot in cattle and swine. After parenteral administration, ceftiofur is rapidly metabolized into desfuroylceftiofur (DFC) and furoic acid through cleavage of its thioester bond (Figure 1a,c). DFC is a metabolite of ceftiofur that can bind to macromolecules, or it is converted to other derivatives such as DFC-cysteine disulfide, DFC-dimer, or a glutathione-conjugate. Nitrocefin, a chromogenic cephalosporin β-lactam is used to detect β-lactamase activity via color change when the compound is hydrolyzed (Figure 1b). Ampicillin, which is extensively used in both medicine and agriculture, is a penicillin-type β-lactam antibiotic like amoxicillin. It differs from penicillin due to the presence of a unique amino group (Figure 1d). These antibiotics are the primary parenteral and *per os* administrated antibiotics in Washington State dairies [32]. Clavulanic acid was also considered, which is commonly co-administered with penicillin family antibiotics to act as a β-lactamase inhibitor that binds irreversibly to some β-lactamases (Figure 1e).

The binding affinity and the catalytic mechanisms of CTX-M-15 for these compounds were studied through isothermal titration calorimetry (ITC) and enzyme kinetics. Through molecular docking and complex crystal structures of the acyl-enzyme intermediates of wild-type CTX-M-15 with the above β-lactam and β-lactamase inhibitor compounds, the molecular interaction and the structural perturbations in the binding-pocket upon complex formation were monitored. In addition, site-directed mutagenesis for the active serine residue of this enzyme was investigated to compare the binding affinity and catalytic activity with those of wild-type enzymes.

## 2. Results

### 2.1. Global Structure of CTX-M-15

The apo- and complex crystal structures of wild-type CTX-M-15 (GenBank accession number AY044436.1) were obtained at high resolution allowing in-depth studies of conformational changes upon substrate binding (Table 1). The overall structure of CTX-M-15 was very similar to previously reported structures (PDB ID: 1ZG4, 1ZG6, 1XPB, 1IYS, 1JWP) [33,34,35,36]. The crystal structure of the apo-form CTX-M-15 was obtained at 1.8 Å resolution with three molecules in the symmetric unit with the unit cell parameters of a = 170.6 Å, b = 50.9 Å, c = 106.8 Å, β = 113.23°, in the monoclinic space group of C2. The structure of CTX-M-15 was comprised of two domains, α-domain and α/β-domain. The α-domain consisted of nine helices (from α-2 to α-10), and the α/β-domain consisted of three helices (α-1, α-11, and α-12), and four β-strands (β-1, β-5, β-6, and β-7) (Figure 2a).

To identify the conformation of the bound β-lactams/inhibitor and their interactions with neighboring residues in the binding pocket, the complex crystals of β-lactams and β-lactamase inhibitor were established by diffusing the compounds to the apo-form crystal. From early stages of crystallographic refinement, the electron density corresponding to each β-lactam and inhibitor compounds was clearly observed, which indicated formation of a covalent bond between the C2 carbonyl of the β-lactam ring and the hydroxyl group of Ser70 as previously indicated [37]. Several different soaking durations were attempted for each β-lactam, but well-defined electron density for β-lactams became diffusive with a lower contour level in either shorter or longer soaking periods. Despite multiple trials, the corresponding electron densities for bound nitrocefin and ceftiofur displayed low contour levels (less than 0.5 sigma). Although formation of the covalent bond between Ser70 and C2 carbonyl of those β-lactams was obvious, their terminal functional groups displayed multiple conformations. Thus, the complex crystal structures of nitrocefin and ceftiofur were not described for this study.

The position of all the atoms in apo- and complex-form with DFC, ampicillin, and clavulanic acid structures were superimposed with RMSD values of 0.23, 0.40, and 0.33Å respectively, reflecting conservation of overall structure upon binding those drugs. However, close inspection of these acyl-enzyme complexes compared with that of the apo-form structure showed significant conformational changes among several residues located in their binding pockets. The binding pocket of CTX-M-15 for those β-lactams was established by the residues from both the α- and α/β-domains, surrounded by the catalytic residues Ser70, Lys73, Ser130, and Glu166 (Figure 2b). The oxyanion hole was clearly observed in all complex structures, which was established by the backbone amide nitrogen atoms of Ser70 and Ser237 with the carbonyl oxygen of the β-lactams and inhibitor present in it.

### 2.2. Clavulanic Acid Complex

Inspection of the electron density map of the complex crystal structure of CTX-M-15 with clavulanic acid (1.4 Å resolution) displayed the cleaved N1–C2 bond of the β-lactam ring and the cleaved C4–O5 bond of the oxazoline ring (Figure 1e). This was consistent for all three clavulanic acid molecules in the three CTX-M-15 molecules of the asymmetric unit. The newly formed imine bond in the opened-ring clavulanic acid was in a *trans* conformation parallelly oriented with the sidechains of Tyr105 and Asn132 but without any significant interaction with those two residues (Figure 3a). The *2Fo-Fc* map of covalently bound clavulanic acid at different counter levels—3.0, 2.0, and 1.0 sigma—are shown in Appendix A. Although association and breakage of clavulanic acid did not seem to produce any major changes on the overall structure of CTX-M-15, close inspection of the superimposed apo- and complex structure showed several subtle but significant structural adjustments (Appendix A). The side chain of Lys73 was shifted toward Ser130, probably accelerating a proton transfer from the sidechain of Ser130 to the nitrogen atom of clavulanic acid. The sidechains of Glu166 and Asn170 also changed their conformation, approaching closer to the oxygen atom of the newly formed acyloxyl group, which resulted in a hydrogen bond network involving Lys73, Glu166, Asn170, and a crystallographic water molecule (Figure 3b). Furthermore, the carboxyl oxygens of clavulanic acid established a hydrogen bond with the sidechain atom of Asn170 and the hydroxyl group of clavulanic acid displayed intramolecular hydrogen bond interactions with its own carbonyl oxygen. The bridged-head nitrogen of clavulanic acid showed a hydrogen bond interaction with the backbone of Ser237. In addition, the association of clavulanic acid displaced three crystallographic water molecules from the active site of the apo-form CTX-M-15. In the complex structure, one crystallographic water molecule was located within a hydrogen bond distance from the carboxyl sidechain of Glu166 and the a-hydrogen of C3. In particular, the sidechain of Ser237 formed a hydrogen bond with the oxygen (O5) of the ring-opened clavulanic acid. The backbone nitrogen atoms of Ser70 and Ser237 created an oxyanion hole via hydrogen bond network with the carbonyl oxygen of acyloxyl group as observed for other acylated β-lactam compounds such as ampicillin and DFC (Figure 4 and Figure 5).

### 2.3. Ampicillin Complex

The corresponding electron density for ampicillin in its complex with CTX-M-15 was well-defined without any discontinuation or breakage at contour level 1.0 sigma (Figure 4a). The *2Fo-Fc* map of covalently bound ampicillin at different counter levels 3.0, 2.0, and 1.0 sigma are shown in Appendix A. In its trans-enamine intermediate, the C2 atom of the lactam ring in ampicillin was covalently bonded to the hydroxyl sidechain of Ser70 (Figure 4b). The backbone amides of Ser70 and Ser237 generated the oxyanion hole that stabilized the position of the negatively charged carbonyl oxygen of the acyloxyl group in the acyl-enzyme intermediate. Upon superimposing the apo- and complex structures, significant conformational changes were observed for sidechains of several residues in the binding pocket, although there was no altered position in the backbone of the enzyme (Appendix A). The ampicillin in its acyl-enzyme intermediate displayed various interactions with the sidechain and backbone atoms of the surrounding residues. The carbonyl oxygen atom of the ampicillin amide group had a hydrogen bond interaction with the oxygen backbone of Ser130. The nitrogen atom of the same amide group in ampicillin was located within a hydrogen bond distance from the hydroxyl sidechain of Ser70. The primary amine in ampicillin, which differentiates it from the parent compound, penicillin, was oriented parallelly with the sidechain of Tyr105. In addition, a parallel p–p stacking interaction was observed between the phenyl ring of Tyr105 and the benzyl ring of ampicillin. Two methyl groups of the penam ring of ampicillin showed hydrophobic interactions with the methyl sidechain of Thr235. The sulfur atom of the same penam ring did not display any specific interaction with neighboring residues, although the bridgehead nitrogen in the penam ring showed hydrogen bond interactions with the backbone and sidechain of Ser237. The carboxyl group attached to the penam ring was within a hydrogen bond distance from the guanidium sidechain of Arg274.

### 2.4. Desfuroylceftiofur (DFC) Complex

The complex crystal structure of CTX-M-15 displayed that the DFC molecule was also covalently bound to Ser70. The electron density map of covalently bound DFC showed at contour level 1.0 sigma (Figure 5a) and contour level 3.0, 2.0, and 1.0 sigma at different orientations in the Appendix A. Upon superimposing the structure of its apo and complex structure, obvious conformational changes were observed for Ser70, Asn104, Asn132, Asn170, Thr216, and Ser237 (Appendix A). The carbonyl oxygen of the acyloxyl group in the acyl-enzyme intermediate was again stabilized by the backbones of Ser70 and Ser237 in the oxyanion hole. The carbonyl oxygen atom of the amide group in DFC was within a hydrogen bond distance from the backbone and sidechain of Ser70, and its nitrogen atom was also within a hydrogen bond distance from the backbone and sidechain of Ser237. In addition, hydrogen bonds were established between the methoxyimine group of DFC and the sidechains of Asn104, Asn132, and Asn170. The aminothiazole ring of DFC was involved in a p–p stacking interaction with the sidechain of Tyr105, but the amine group on the same ring did not show any significant interaction. The thiol group attached to the cephem ring showed a hypervalent interaction with the sidechain of Thr216 [38]. The bridgehead nitrogen in the cephem ring showed a polar interaction with the sidechain of Ser237, and the sulfur atom in this ring showed interaction with the sidechain of Ser130.

### 2.5. Steady-State Kinetics of CTX-M-15

The enzymatic activities of the wild-type and mutant CTX-M-15 were assayed for the four β-lactam compounds. A range of concentrations (0–3 mM) for the β-lactams were tested and the enzyme concentration was maintained at 10 nM. The steady-state enzyme kinetics of wild-type CTX-M-15 for all four β-lactam compounds showed a typical Michalis–Menten type curve (Appendix A). The corresponding V_max_, K_m_, k_cat_ for the four β-lactams are found in Table 2. The K_m_ values displayed an increasing trend in the order of DFC, ampicillin, ceftiofur, and nitrocefin. Overall, CTX-M-15 showed higher catalytic efficiency for DFC and nitrocefin than ampicillin and ceftiofur.

To confirm the potency of Ser70 as a nucleophilic residue in the hydrolysis reaction of the CTX-M-15, the corresponding serine residue was mutated to alanine (S70A) and was purified to the same purity level as the of wild-type enzyme. Upon mutation, the K_m_ for those compounds were substantially reduced (Table 2) and the calculated k_cat_/K_m_ for DFC, ampicillin, ceftiofur, and nitrocefin decreased 43-, 17-, 12-, and 46-fold compared to those of wildtype CTX-M-15 confirming the catalytic role of Ser70.

Inhibition of the clavulanic acid has been measured for CTX-M-15 with ceftiofur as a substrate, which showed a mixed inhibition affecting both K_m_ and V_max_ (Appendix A). The calculated values for K_i_ were 0.022 (±0.007) μM. Considering α value 2.28 (±0.01), clavulanic acid preferentially binds to the free enzyme.

### 2.6. Isothermal Titration Calorimetry

To analyze the thermodynamic parameters of the interactions between CTX-M-15 enzyme and β-lactam compounds, isothermal titration calorimetry (ITC) was applied using the S70A-CTX-M-15 mutant enzyme (Table 3). As expected, binding of all four β-lactam compounds and lactamase inhibitor displayed a negative Gibbs free energy change (ΔG); −6.7, −6.2, −6.0, −5.7, and −4.8 kcal mol^−1^ for DFC, ampicillin, ceftiofur, nitrocefin, and clavulanic acid, respectively. The corresponding enthalpy changes (ΔH) were −2247, −509, −1157, −2025, and −1333 cal mol^−1^ for DFC, ampicillin, ceftiofur, nitrocefin, and clavulanic acid respectively. The entropic changes (ΔS) were 15.2, 19.1, 16.3, 12.1, and 11.7 cal mol^−1^ K^−1^ for DFC, ampicillin, ceftiofur, nitrocefin, and clavulanic acid, respectively. Thus, association of all five β-lactams to CTX-M-15 were driven by both enthalpically and entropically. The calculated k_d_ values for DFC, ampicillin, ceftiofur, nitrocefin, and clavulanic acid were 10.7, 28.8, 38.2, 71.3, and 288.6 μM, respectively (Appendix A).

### 2.7. Molecular Docking

Dominant heat of the catalytic reaction prevented analysis of wild-type enzymes using ITC for the heat of association, so a molecular docking approach was used to confirm and validate the thermodynamic parameters obtained from mutant ITC (Table 4). Utilizing AutoDock Vina, the compounds were docked into the crystal structure of the wild-type enzyme. Additionally, the compounds were docked into the S70A-CTX-M-15 mutant structure that Ser70 was mutated to Ala70 from the wild type CTX-M-15 crystal structure. The ΔG for WT-CTX-M-15 with DFC, ampicillin, ceftiofur, nitrocefin, and clavulanic acid were calculated as −7.5, −6.9, −6.8, −6.7, and −6.2 kcal mol^−1^, respectively. The S70A-CTX-M-15 ΔG values for docking were −8.6, −7.4, −7.6, −7.4, and −6.0 kcal mol^−1^ for DFC, ampicillin, ceftiofur, nitrocefin, and clavulanic acid, respectively. These ΔG values of wild type and mutant from molecular docking displayed a similar trend as observed ΔG values from ITC.

As expected, most of the conformers located in the putative antibiotic-binding cleft with their cleavable lactam ring near the Ser70 sidechain were similar to those in complex crystal structures. The pose of the lowest energy conformation was selected and analyzed for interactions with the residues in the binding pocket. A docked ampicillin molecule showed hydrogen bonds and salt bridge interactions with the sidechains of Ser70, Ser130, Thr216, Arg274, and Thr235. In specific, the amine group established a hydrogen bond interaction with the backbone and sidechain of Ser70, while the β-lactam ring shifted toward the sidechains of Ser237 and Thr235 without any significant interactions. The penam ring of ampicillin was located between the sidechain of Thr216 and Arg274, and its carboxyl group displayed a hydrogen bond interaction with the sidechains of Thr216 and a salt bridge interaction with the sidechain of Arg274. The carbonyl oxygen of the amide group displayed hydrogen bonds with the sidechains of Ser70, Ser130, and Thr235. The benzene ring of ampicillin was oriented vertically to the phenolic sidechain of Tyr105, showing a cation-π interaction (Figure 6a).

DFC also interacted with the backbones and sidechains of Ser70, Asn104, Ser130, Thr216, Lys234, Thr235, and Ser237. The β-lactam ring of DFC showed hydrogen bond interactions with the sidechains of Ser70 and Ser130. The cephem ring was oriented face-to-face stacked with the phenolic sidechain of Tyr105 and its carboxyl group had hydrogen bond interactions with the sidechains of Ser130, Thr235, and Ser237, and also a salt bridge interaction with the amine sidechain of Lys234. In addition, its thiol group displayed a hypervalent interaction with the hydroxyl sidechain of Thr216. The nitrogen atom of the amide group showed a hydrogen bond interaction with the sidechain of Ser237. Both the nitrogen atom and the amine substituent of the aminothiazole ring showed hydrogen bond interactions with the sidechain of Asn104. The methoxyimine parallelly oriented with the backbone of Gly236 without any obvious interactions (Figure 6b).

Nitrocefin showed polar interactions with the backbones and sidechains of Asn104, Tyr105, Ser130, Asn132, and Asn170. The β-lactam ring displayed a hydrogen bond interaction with the sidechain of Asn170. The cephem ring was parallelly oriented with the phenolic ring of Tyr105, and its carboxyl group had hydrogen bond interactions with the backbone and sidechains of Ser130 and with the sidechain of Asn132. The nitrogen atom of amide group showed a hydrogen bond interaction with the sidechain of Asn170. The sulfur atom of the thiophene ring displayed a hypervalent interaction with the sidechain of Asn104. The para-nitro group of dinitrobenzene ring showed a hydrogen bond interaction with the phenolic hydroxyl of sidechain of Tyr105 (Appendix A).

Ceftiofur interacted with the backbones and sidechains of Ser70, Asn104, Ser130, Lys234, Thr235, and Ser237. The b-lactam ring of ceftiofur showed hydrogen bond interactions with the sidechain of Ser70 and Ser130. As in nitrocefin, the cephem ring was also parallelly oriented to the phenolic sidechain of Tyr105. Its carboxyl group formed hydrogen bond interactions with the backbone of Ser237 and the sidechains of Ser130 and Thr235, and a salt bridge interaction with the sidechain of Lys234. The carbonyl of the amide group had a hydrogen bond interaction with the sidechain of Asn104. The oxygen atom of furfural ring displayed a hydrogen bond interaction with the hydroxyl sidechain of Ser237. The aminothiazole ring and methoxyimine did not show any specific interactions (Appendix A).

## 3. Discussion

With the emergence of pathogenic bacteria strains that exhibit broad spectrum antibiotic resistance, CTX-M enzymes have become a dominant gene in most ESBLs in the world [39]. Consistent with this dominance is the recent ascendance of *bla*_CTX-M-15 -_positive strains of *E. coli* found in dairy cattle in Washington State [31]. Cephalosporins and penicillins have been popular antibiotics, often being co-administrated with inhibitors such as clavulanic acid to treat a number of food-animal diseases. Ceftiofur, DFC, and nitrocefin share a core structure made of β-lactam, cephem rings, and a carboxyl substituent on the cephem ring. Ceftiofur contains a thiol-furfural substituent on the cephem ring, and its C3 atom on the β-lactam ring has an amide substituent composed of an aminothiazole ring and methoxyimine group (Figure 1a). DFC, a metabolite of ceftiofur, does not have furfural substituent (Figure 1c). Nitrocefin has a *meta*-dinitrobenzene alkene substituent on the cephem ring, and a thiophene amide substituent on the C3 atom of the β-lactam ring (Figure 1b). Ampicillin is composed of β-lactam and penam rings, with two methyl groups and a carboxyl substituent on the penam ring. It also contains an amide group with a benzyl and primary amine substituent on the C3 atom of the β-lactam ring (Figure 1d). Clavulanic acid has β-lactam and oxazoline rings. Its substituents are composed of a carboxylate and a hydroxyl alkyl on the oxazoline ring (Figure 1e).

### 3.1. Structural, Kinetic, and Thermodynamic Analysis for Complex Formation

Complex crystal structures and molecular docked structures from this work indicated that substrate-binding was established through polar and hydrogen bond interactions with surrounding residues in the active site pocket. Residues in both the α-domain and α/β-domain of CTX-M-15 participated in catalyzing and binding β-lactam compounds, which included Ser70, Lys73, Asn104, Tyr105, Ser130, Asn132, Glu166, Asn170, Thr216, Ser237, and Arg274. The number of interactions with neighboring residues increased with the molecular mass of the tested compounds clavulanic acid (199 Da) < DFC (322 Da) < ampicillin (349 Da) < nitrocefin (516 Da) < ceftiofur (545 Da). DFC and ampicillin interacted with eleven residues, while clavulanic acid interacted with only eight residues. Ceftiofur and nitrocefin interacted with 13 and 12 residues, respectively. Among these interacting residues, five residues were common in all three compounds: Ser70, Lys73, Glu166, Ser130, and Ser237. The crystallographic B-factor of the sidechain for these residues were substantially reduced in the complex crystal structure compared to those in the apo-form crystal structure, indicating their contribution was through an induced fit mechanism. It is likely that the flexibility of these residues allowed the CTX-M-15 enzyme to accommodate various β-lactam molecules.

From the complex crystal structure and activity assays, it is clear that Ser70 served as a nucleophile in covalent catalysis as previously found [37]. Additionally, the structure revealed that the carboxyl sidechain of Glu166 acted as an acid-base catalyst, and the Lys73 and Ser130, seemed to serve in the proton shuttling process. In complex structures of five compounds, Tyr105, Ser130, and Ser237 were commonly involved in capturing substrates. Hydrogen bond interactions were observed in all compounds between the amide group of β-lactam compounds and the backbone/sidechain of both Ser130 and Ser237. Several unique interactions were also observed in each compound as described in the result section.

It is apparent that hydrolysis of the mutants are minuscule (Figure 6). For all substrates tested, the turnover numbers are two to three orders of magnitude lower than the wildtype. Some minor hydrolytic activity still existed probably due to other S70-independent mechanism (covalent catalysis). One of the plausible mechanisms is that a nucleophilic attack may still proceed with a water molecule to produce a tetrahedral intermediate followed by a collapse to yield the hydrolyzed product as reported before for the serine protease [40]. In addition, preferential binding of the transition state, such as using an oxyanionic hole, could increase a local concentration of transition state as shown in general catalytic antibodies [41].

The turnover number, k_cat_ of CTX-M-15 for DFC, nitrocefin, ceftiofur, and ampicillin displayed in decreasing order, 489.8, 338.3, 313.7, and 101.9 s^−1^, respectively (Table 2). The mutant enzyme, S70A-CTX-M-15, revealed a significant reduction in k_cat_ to all compounds confirming Ser70 as a key residue in its acylation reaction. In addition, the ITC data with S70A-CTX-M-15, together with molecular docking results confirmed that a magnitude of dissociation constant, k_d_, toward all compounds followed a similar trend with regard to K_m_ values in the steady-state kinetics. The k_d_ values were increased in the order of DFC (10.7 μM), ampicillin (28.8 μM), ceftiofur (38.2 μM), nitrocefin (71.3 μM), and clavulanic acid (288.6 μM). ITC data indicated that all binding reactions were driven both enthalpically and entropically. In particular, binding of nitrocefin and DFC had larger magnitude of negative ΔH, indicating that a notable amount of bonding interaction would be generated. In addition, the crystal structure of the apo-form CTX-M-15 indicated that there were several water molecules in the binding pocket. Thus, association of compounds would gain entropic contribution by replacing these molecules.

Significantly, ampicillin displayed the largest entropy, 19.1 cal mol^−1^ K^−1^, which probably indicates that bound ampicillin and/or interacting residues of CTX-M-15 maintain higher mobility in the binding pocket than other tested compounds. Ampicillin also displayed the lowest catalytic efficiency, 0.52 μM^−1^ s^−1^ among four lactams. Although its ΔH of association was lowest (−509 cal mol^−1^) for ampicillin, probably due to its relatively smaller size than other compounds, the overall ΔG value was similar to others, indicating compensation with higher entropic gain. Thus, ampicillin could effectively inhibit the lactamase action in bacteria. On the contrary, DFC, which is a similar molecular mass and binding affinity as ampicillin, was more effectively destroyed by CTX-M-15 (3.46 μM^−1^ s^−1^) compared to ampicillin (0.52 μM^−1^ s^−1^). To understand this low efficiency against ampicillin, molecular interactions between amino acid residues in the binding pocket and the functional groups of ampicillin and DFC were compared.

There were significant differences for the association between CTX-M-15 and DFC or ampicillin. One was the orientation of the β-lactam ring at the binding pocket. The docked β-lactam ring of DFC established hydrogen bond interactions with the sidechains of Ser70 and Ser130, whereas the β-lactam ring of ampicillin shifted toward the sidechain of Ser237. Another difference was the interaction of the carboxyl group on the penam ring of ampicillin and the cephem ring of DFC. The ampicillin’s carboxyl group established a salt bridge with the guanidinium sidechain of Arg274 and a hydrogen bond with the hydroxyl sidechain of Thr216, while the DFC’s carboxyl group established hydrogen bonds with the hydroxyl sidechains of Ser130 and Ser237, Thr235, and a salt bridge interaction with the amine sidechain of Lys234. Overall, DFC had significantly more interactions with the residues in the binding pocket compared to ampicillin, confirming the magnitude of its enthalpy (Table 3). Upon amine group interaction with sidechain of Ser70, the cleavable bond of ampicillin’s β-lactam ring was located slightly away from the hydroxyl sidechain of Ser70. We surmise that these unique interactions could position the ampicillin lactam ring in a non-ideal position for acylation and diacylation reaction by Ser70, consistent with the reduced catalytic efficiency of CTX-M-15 towards ampicillin decreases.

Ceftiofur is used to treat pneumonia, metritis, and foot rot in dairy cattle [42,43,44]. After administration, ceftiofur undergoes rapid conversion into desfuroylceftiofur (DFC) that undergoes reversible conversion primarily into DFC-cysteine, and DFC-dimer [45,46]. The extra furfural moiety might make ceftiofur a superior antibiotic compared to DFC (Table 2). The ΔH for ceftiofur’s binding reaction is half of DFC’s, which reflects its diminished molecular interaction. Despite the flexible nature indicated in the complex crystal structure, the furfural moiety in the molecular docked structure displayed a π-stacking interaction with the sidechain of Tyr105. In addition, a hydrogen bond interaction existed between the carbonyl oxygen of the furfural group and the sidechain of Ser237. The aminothiazole ring of DFC showed a π-stacking interaction with the phenolic sidechain of Tyr105, while the same ring in ceftiofur did not have any significant interaction with the residues in enzyme pocket. In addition, the DFC’s methoxyimine group established hydrogen bond interactions with three asparagine amino acids (Asn104, Asn132, and Asn170) via parallel orientation of its aminothiazole ring with the sidechain of Tyr105, whereas ceftiofur’s methoxyimine group shifted from the sidechain of Ser237 via its furfural group in parallel orientation to the sidechain of Tyr105. The small negative value of enthalpy (ΔH = −1157 cal mol^−1^) confirmed the weak interaction of ceftiofur with the active site of CTX-M-15. Probably these furfural moiety-mediated interactions could put the ceftiofur’s cleavable lactam ring into non-ideal position/orientation at the active site pocket. Thus, absence of the furfural group as in DFC caused higher ΔH (−2247 cal mol^−1^) and higher catalytic activity (3.46 μM^−1^ s^−1^). The large negative value of enthalpy could allow for effective interactions between the DFC and catalytic site of the enzyme.

### 3.2. Enzyme Catalytic Mechanism and Inhibition Mechanism

CTX-M-15, which belongs to hydrolases (EC 3.5.2.6), cleaves lactam carbon–nitrogen amide bonds rapidly, deactivating β-lactam antibiotics [47,48]. In general, this β-lactam hydrolysis proceeds through an acylation-diacylation reaction. During acylation, one acidic residue acts as a general-base catalyzing serine residue’s nucleophilic attack of the lactam carbonyl carbon and formation of a tetrahedral intermediate. Collapse of the tetrahedral intermediate proceeds to the oxyanion-hole stabilized acyl-enzyme adduct. Diacylation proceeds as the glutamic acid residue acts as a general-base and catalyzes a hydrolytic water attack on the carbonyl carbon of the adduct, resulting in the release of the hydrolyzed β-lactam and return of CTX-M-15 to its resting state.

#### 3.2.1. Hydrolysis Mechanism

From the complex crystal structures of the three compounds—clavulanic acid, ampicillin, and DFC—it is clear that the nucleophilicity of Ser70 in CTX-M-15 is boosted through an established hydrogen bond network with the neighboring residues—Glu166, Lys73, and Ser130. In all complex structures, the position of the carboxyl sidechain in Glu166 is altered upon complex formation, thus enabling proton abstraction from the hydroxyl sidechain of Ser70 acting as a general base. The deprotonated sidechain of Ser70 forms the tetrahedral transition state of the acyl-enzyme. Meanwhile, the backbone amide groups of both Ser70 and Ser237 stabilize the negative charge of the resulting alkoxide ion through the preexisting oxyanion hole. The proton is shuttled to the nitrogen atom of the β-lactam via the sidechains of Ser130, Lys73, and Glu166 through the hydrogen bond network. The hydrolysis mechanisms of DFC and ampicillin were illustrated in Appendix A, respectively. Ceftiofur and nitrocefin should follow the same mechanism.

#### 3.2.2. Inhibition Mechanism

Based on the structural data, an inhibition mechanism by clavulanic acid could be due to first cleaving the b-lactam ring and then opening of the oxazoline ring. The first step is the same reaction seen for all four b-lactam compounds and involves formation of an acyl-enzyme intermediate. This is followed by the protonation of the C6–C9 double bond by the nearby crystallographic water molecular. This water molecule is located at an appropriate distance not only to donate a proton to the C6–C9 double bond, but to abstract a proton from the amine sidechain of Lys73. In addition, the Lys73 was properly located to abstract a proton from the bridgehead nitrogen opening the oxazoline ring. The newly formed carbonyl group of the cleaved oxazoline ring could be stabilized by the hydroxyl sidechain of Ser237 through hydrogen bond formation. The position of C3 atom in trans-imine intermediate is located next to the carbonyl group. The a-proton of C3 is abstracted by the same water molecule together with the proton network established with the sidechains of Glu166 and Lys73. The pKa of this C3 a-hydrogen could be low, because the resulting negative charge was delocalized through a double bond formation (C2 = C3) and the carbonyl oxygen in acyloxyl group. In addition, the stabilization of negative charge of the carbonyl oxygen would be accomplished by the existing oxyanion hole made by Ser237 and Ser70, maintaining a double-bond between C2 and C3 thus preventing hydrolysis (Figure 7).

Collectively, the crystal structure evidence presented herein demonstrates important differences in how CTX-M-15 processes β-lactam antibiotics. These comparative findings indicate that: (1) compounds that incorporate an amine moiety (e.g., ampicillin) or a furfural moiety (e.g., ceftiofur) are likely to exhibit greater resistance to the hydrolytic activity of CTX-M-15; and (2) the understanding that inactivation of CTX-M-15 by clavulanic acid involves nucleophilic Ser70 may aid in the design of future β-lactamase inactivators to overcome for β-lactam-resistant bacteria.

## 4. Materials and Methods

### 4.1. Accession Numbers

Sequence data from this article can be found in the EMBL/GenBank data libraries under accession number (*AY044436.1*), and the structures discussed in this manuscript can be found at www.rcsb.org (accessed on 1 March 2022) deposited under the PDB ID: 7U48, 7U4B, 7U49, and 7U57.

### 4.2. Chemicals and General Reagents

Analytical-grade chemicals were obtained from Sigma-Aldrich (St. Louis, MO, USA), Thermo Fisher (Waltham, MA, USA) and Alfa-Aesar (Ward Hill, MA, USA). Screening solutions for crystallization were obtained from Hampton Research (Aliso Viejo, CA, USA). Unless otherwise noted, basic reagents and antibiotics were purchased from Sigma-Aldrich (St. Louis, MO, USA), VWR International (Radnor, PA, USA), and LLC (Largo, FL, USA), respectively. Molecular graphics images were produced using the Chimera package (UCSF, NIH P41 RR-01081, San Francisco, CA, USA). The plot figures were generated by GraphPad Prism (San Diego, CA, USA). The molecule figures were built by the MarvinSketch v.17.28.0. (Budapest, Hungary).

### 4.3. Expression and Purification of Recombinant Protein

Wild-type CTX-M-15 was expressed after inserting *bla*_CTX-M-15_ into a high-copy number pET200 TOPO vector under the control of a T7lac promoter for IPTG-inducible expression, which that was then transformed into *E. coli,* as previously described [37]. Clone BL21(DE3)/pET200-CTX-M-15 was grown in 3L of Luria–Bertani medium containing 50 μg mL^−1^ kanamycin at 37 °C until cultures reached OD_600_ of 0.5. Protein production was induced by adding 1.0 mM isopropyl β-thio-galactopyranoside (IPTG) at 30 °C for 18 h. The harvested cells were resuspended in 30 mL of 20 mM Tris Buffer, pH 8.0 and sonicated by sonifier (Model 450; Branson Ultrasonics, Danbury, CT, USA). After centrifugation at 17,000 rpm for 1 h, the supernatant was loaded at 7 mL/min on an anion-exchange column Hiprep DEAE Sepharose 16/10 (GE Healthcare), which was previously equilibrated with 20 mM Tris buffer, pH 8.0. The protein eluent was collected in the flowthrough and dialyzed in 5 mM potassium phosphate buffer, pH 6.8 overnight. The enzyme was concentrated using Amicon Ultra 15-10 K (Sigma-Aldrich, St. Louis, MO, USA) down to 10 mL and applied to a pre-equilibrated hydroxyapatite column (Bio-Rad, Hercules, CA, USA) with 5 mM potassium phosphate buffer, pH 6.8. The protein was eluted by using a linear potassium phosphate gradient at 3 mL/min in a 50 mM potassium phosphate fraction. The purity of the enzyme was analyzed by SDS-PAGE in all expression and purification steps, and the protein concentration (10 mg/mL) was determined by Bradford assay (Bio-Rad, Hercules, CA, USA).

### 4.4. Crystallization

Crystals of CTX-M-15 were obtained by vapor diffusion in a hanging-drop setup. Protein was concentrated to 10 mg/mL in 50 mM potassium phosphate at pH 6.8 by using an Amicon 8050 ultrafiltration cell with a 10-kDa cutoff membrane (EMDMillipore, St. Louis, MO, USA). A commercial crystallization kit, Index (hamptonresearch, Aliso Viejo, CA, USA), was used for crystal screening through the sitting-drop, vapor-diffusion method by Crystal Phoenix (Art Robbins Instruments, Sunnyvale, CA, USA). Initial crystals appeared in the solution of 300 mM ammonium sulfate and 30% PEG 4000 at 20 °C. With the same condition, larger crystals were reproduced by hanging-drop vapor-diffusion method. The crystallization drops included 2 μL of enzyme (5 to 10 mg/mL) and 2 μL of crystallization reagent (300 mM ammonium sulfate and 30% PEG 4000) and were equilibrated with the same solution of 500 μL. Cubic-shaped apo crystals started to appear after three days. Complex crystals were made by soaking apo-enzyme crystals with each antibiotic compound; ceftiofur, desfuroylceftiofur (DFC), and ampicillin, one chromogenic cephalosporin substrate nitrocefin, and one inhibitor clavulanic acid. All complex structures were made by soaking apo-crystals into the same crystallization mother liquor solution containing 10 mg/mL of each compound for 10 min. Soaking time of clavulanic acid was empirically adjusted to 30 min. Crystals were then mounted by CryoLoop and frozen immediately by dipping them into liquid nitrogen, and their diffraction data were collected.

### 4.5. Structure Determination

Crystallographic diffraction data were collected at the Advanced Light Source (ALS) Beamline 5.0.2 at 100K and scaled up by using HKL2000. The structure of native CTX-M-15 (PDB ID:4HBT) [37] was used as an input model for molecular replacement in PHENIX Phaser (PHENIX Industrial Consortium; Berkeley, CA, USA) [49]. After generating primary phases from the model, it was appropriately fitted to the electron density by applying COOT (Biomedical Campus, Cambridge, UK) [50]. Refinement of the enzyme was set up in PHENIX and the substrate was built by running ELBOW. The 3D structure of the substrate was merged in the refined model using COOT, and ReadySet was completed in PHENIX to build the complex structure. Further cycle refinements were performed until the substrate was well-fitted in the electron density. Structure factors and crystallographic coordinates have been deposited in the in the Research Collaboratory for Structural Bioinformatics Protein Data Bank (RCSB PDB) with PDB IDs of 7U57, 7U48, 7U4B, and 7U49 for the apo-, clavulanic acid-, ampicillin-, and DFC-WT-CTX-M-15, respectively for the C121 space group. The final R_work_ was 21.19%, 18.70%, and 24.00%, respectively, R_free_ was 26.99%, 25.08%, and 31.62% for clavulanic acid-, ampicillin-, and DFC-WT-CTX-M-15, respectively. The statistics for the diffraction data are listed in Table 1.

### 4.6. Site-Directed Mutagenesis

To mutate the serine located at position 70, we performed site-directed mutagenesis using the QuikChange II Site-Directed Mutagenesis kit (Agilent Technologies, La Jolla, CA, USA) following the manufacturer’s instructions. Briefly, plasmid DNA was extracted from overnight cultures using PureLink^TM^ Quick Plasmid Miniprep kit (Invitrogen) following manufacturer’s instructions. *PfuUltra* HF DNA polymerase was used to generate PCR amplicons containing the desired mutation for each gene. PCR conditions included 95 °C for 30 s and then 95 °C (30 s), 55 °C (1 m), and 68 °C (6 m) for 16 cycles. *Dpn I* digestion was performed on resulting amplicons. Plasmids were then transformed into XL1-Blue Super competent cells by heat shock method. Transformants were selected on LB agar containing carbenicillin (100 mg/mL). Mutants were confirmed by sequencing both strands using T7 and T7term primers. The CTX-M-15—“AGC” was mutated for “GTG”.

### 4.7. Steady-State Kinetics Experiment

Enzymatic activities of wild type and mutant CTX-M-15 were measured with ampicillin, ceftiofur, DFC, and nitrocefin. The concentration of each β-lactam were ranged from 0–3 mM in PBS (157 mM Na^+^, 140 mM Cl^−^, 4.45 mM K^+^, 10.1 mM HPO_4_^2−^, 1.76 mM H_2_PO_4_^−^) that is pH 7.4. The reaction was initiated by adding 75 μL of stock solutions of wildtype and mutant enzymes (40 nM) to 225 μL reaction buffer resulting in a final enzyme concentration of 10 nM. The enzyme concentration of the stock solution was measured by Bradford assay. The absorbance change was monitored at the specific wavelength for each substrate and hydrolyzed substrate (289 nm for ceftiofur, and 238 nm for hydrolyzed ceftiofur; 260 nm for DFC, and 225 nm for hydrolyzed DFC; and 220 nm for ampicillin, and 212 nm for hydrolyzed ampicillin; 486 nm for nitrocefin) over 10 s intervals for 30 min using a 96-well plate in triplicate. The inhibition action of clavulanic acid was measured by monitoring the hydrolysis of ceftiofur (0–1500 μM) by 10 nM CTX-M-15 in the presence of different amount of clavulanic acid (0–50 μM) in PBS buffer, pH 7.4 at 37 °C. Changes in absorbance were monitored using Spark multimode microplate reader (Tecan Trading AG, Switzerland). The rate of the hydrolyzed product was calculated by the Beer–Lambert law and then using the software of Prism GraphPad to determine kinetic parameters *k_cat_*/*K_m_*, *k_cat_, K_m_*, *V_max_*, and *K_i_*.

### 4.8. Isothermal Calorimetry Titrations (ITC)

Isothermal calorimetry titrations were performed for S70A-CTX-M-15 and β-lactams in MicroCal -PEAQ-ITC instrument (Malvern Panalytical, Malvern, UK). Enzyme was prepared in 100 mM phosphate buffer, pH 7.4. The concentration of enzyme in the calorimetric titration cell was diluted to 75 μM. All titrations were performed at 25 °C with a stirring speed of 750 rpm with 27 injections (1.4 mL each). Substrates were brought to a concentration of 1.5 mM in the titration buffer and injected into the enzyme solution, and the heat of binding was recorded.

### 4.9. Molecular Docking

Ampicillin, ceftiofur, clavulanic acid, DFC, and nitrocefin were all docked into CTX-M-15 using AutoDock Vina [51]. Preparation of the grid and ligands were completed in AutoDock Tools. S70A-CTX-M-15 mutation was completed in Wincoot using the mutation tool and prepared for docking using AutoDock Tools. All ligands were docked into both the WT and mutated enzyme. CTX-M-15’s clavulanic acid binding site was found by blind docking clavulanic acid into a whole-protein search grid. Upon elucidation of the region with the lowest energy binding region, a 30 Å, 30 Å, 30 Å grid was established centered at the coordinates −48.89 Å, 16.844 Å, 34.551 Å. Ampicillin, DFC, ceftiofur, and nitrocefin were docked using the same procedure as clavulanic acid.

## Figures and Tables

**Figure 1 ijms-23-05229-f001:**
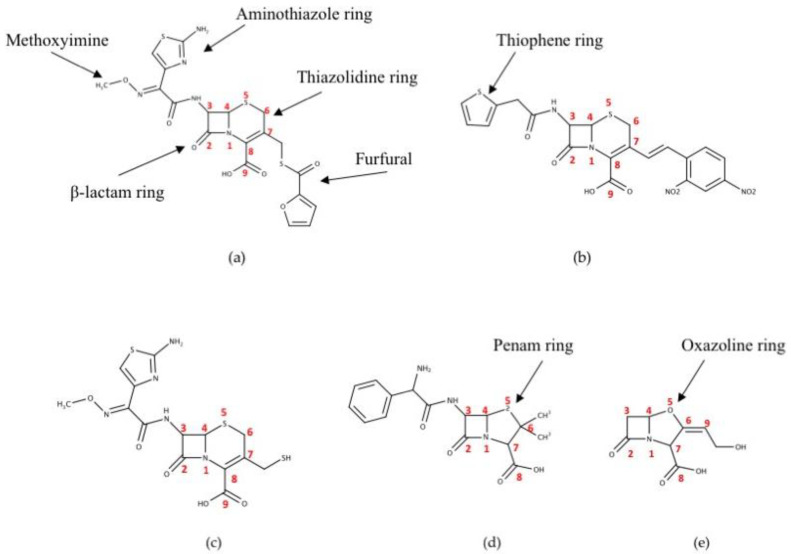
Chemical structures of five β-lactam compounds. (**a**) Ceftiofur; (**b**) Nitrocefin; (**c**) Desfuroylceftiofur (DFC); (**d**) Ampicillin; (**e**) Clavulanic acid. Thiazolidine, β-lactam, aminothiazole, thiophene, penam, and oxazoline rings—and also methoxyimine and furfural groups—are shown with arrows. The figures were generated with the MarvinSketch v.17.28.0. (Budapest, Hungary).

**Figure 2 ijms-23-05229-f002:**
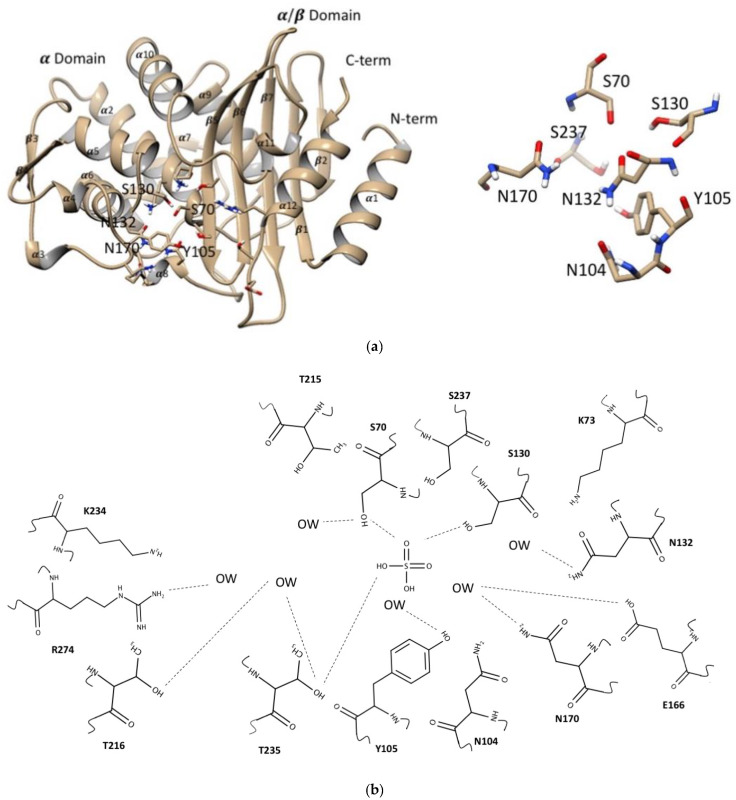
Global structure of CTX-M-15. (**a**) CTX-M-15 is shown with two domains, α and α/β. The active site is positioned toward the center, showing the catalytic Ser70, Lys73, Ser130, and Glu166. The figure was generated with UCSF Chimera v1.13.1. (San Francisco, CA, USA). (**b**) The 2D structure of binding pocket of CTX-M-15 contained the residues and water molecules was built by the MarvinSketch v.17.28.0. (Budapest, Hungary).

**Figure 3 ijms-23-05229-f003:**
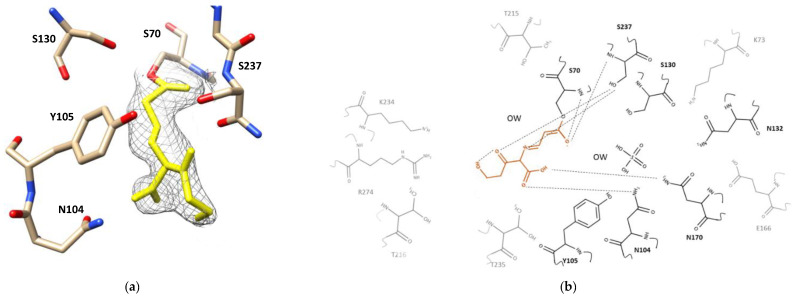
Active site of CTX-M-15 Clavulanic acid-enzyme complex. (**a**) Electron density map of CTX-M-15 active site bound to clavulanic acid shown in yellow. *2Fo-Fc* map (contoured at 1.0 sigma) is shown in grey which was built by UCSF Chimera v1.13.1 (San Francisco, CA, USA). Residues around the binding pocket of the clavulanic acid-CTX-M-15 crystal structure are depicted as cream, red, blue sticks. (**b**) 2D structure of clavulanic acid (orange) in the binding pocket of CTX-M-15. All residues have been shown, hydrogen bonds are drawn using dashed lines. The residues that interacted with clavulanic acid are in dark black, residues without interaction are shown in gray. The figure was built by the MarvinSketch v.17.28.0. (Budapest, Hungary).

**Figure 4 ijms-23-05229-f004:**
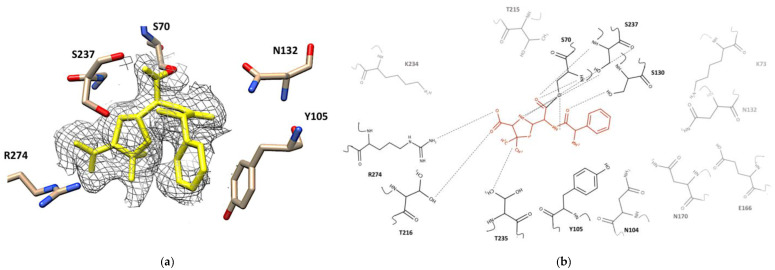
Active site of CTX-M-15 of the ampicillin-enzyme complex. (**a**) Electron density map of the CTX-M-15 active site bound to ampicillin in the yellow. *2Fo-Fc* map (contoured at 1.0 sigma) is shown in dark gray. Residues around the binding pocket of the ampicillin -CTX-M-15 crystal structure are depicted as cream, red, blue sticks. The figure was generated with UCSF Chimera v1.13.1. (San Francisco, CA, USA). (**b**) 2D structure of ampicillin (orange) in the binding pocket of CTX-M-15. All residues have been shown, hydrogen bonds are drawn using dashed lines. The residues that interacted with ampicillin are shown in dark black, residues without interaction are depicted in gray. The figure was built by the MarvinSketch v.17.28.0. (Budapest, Hungary).

**Figure 5 ijms-23-05229-f005:**
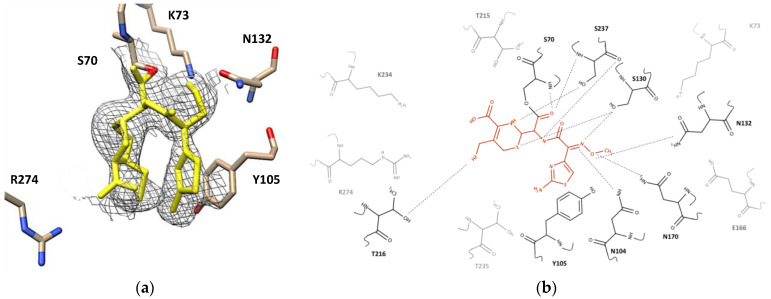
Active site of CTX-M-15 in DFC-enzyme complex. (**a**) Electron density map of the CTX-M-15 active site bound to DFC in the yellow color. The *2Fo-Fc* map (contoured at 1.0 sigma) is shown in gray. Residues around the binding pocket of the DFC-CTX-M-15 crystal structure are depicted as cream, red, blue sticks. The UCSF Chimera v1.13.1. (San Francisco, CA, USA). built this figure. (**b**) The 2D structure of DFC (red color) in the binding pocket of CTX-M-15. All residues have been showed, the hydrogen bond drown by dashed lines. The residues had interaction with DFC were in dark black color, and other ones did not have interaction in the gray color. The figure was generated with the MarvinSketch v.17.28.0. (Budapest, Hungary).

**Figure 6 ijms-23-05229-f006:**
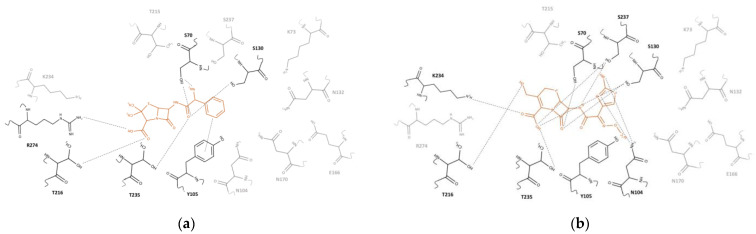
Ligand binding pocket of CTX-M-15. (**a**) Ampicillin is in the binding pocket of enzyme. (**b**) DFC is in the binding pocket of enzyme. Hydrogen bonds are depicted by block dotted lines and the ligand is in the red color. The figures were generated with the MarvinSketch v.17.28.0. (Budapest, Hungary).

**Figure 7 ijms-23-05229-f007:**
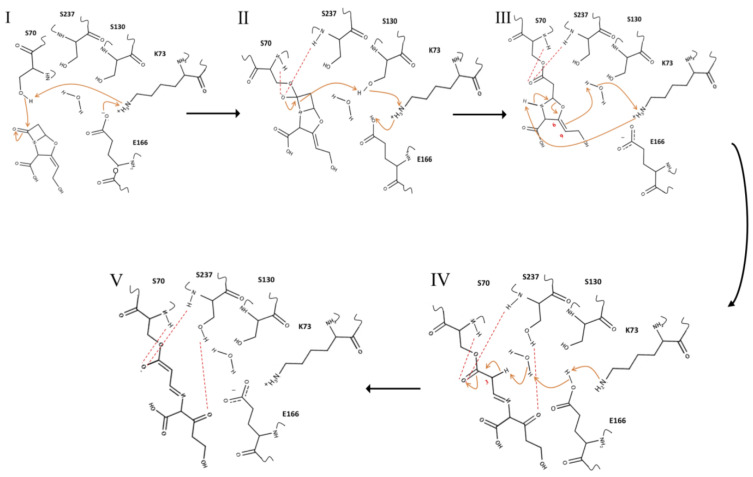
The proposed mechanism for clavulanic acid inhibition of CTX-M-15. Orange colored lines indicate electron transfer and red dashed lines indicate hydrogen bond interactions. Figures were generated using MarvinSketch v.17.28.0. (Budapest, Hungary). (**I**) Clavulanic acid binding is displayed as a series proton and electron transfers. Lys73 is deprotonated by Glu166 and then the neutral Lys73 activates Ser70 by abstracting the proton from its sidechain. The nucleophilic Ser70 attacks the b-lactam ring. (**II**) The tetrahedral transition state is stabilized by an oxyanion hole generated with backbones of Ser70 and Ser237. The bridgehead nitrogen atom abstracts a proton from Ser130, and the shared proton is transferred to Ser130 from Lys73, and then to Lys73 from Glu166. (**III**) The double bond (C6–C9) in the acyl-enzyme intermediate abstracts a proton from a water molecule. The cleavage of the oxazoline ring is followed by abstracting a proton from a water molecule that is protonated by Lys73, and then neutral Lys73 abstracts a proton from the bridgehead nitrogen atom. (**IV**) The negatively charged Glu166 abstracts a proton from a water molecule, the activated water molecule removes an acidic hydrogen on the a-carbon (C3). (**V**) *trans*-imine-enzyme is stabilized by several hydrogen bond interactions.

**Table 1 ijms-23-05229-t001:** Data collection and refinement statistics.

	Apo-CTX-M-15 (7U57)	Clavulanic Acid-CTX-M-15 (7U48)	Ampicillin-CTX-M-15 (7U4B)	DFC-CTX-M-15 (7U49)
Data collection				
Space group	C 1 2 1	C 1 2 1	C 1 2 1	C 1 2 1
Cell dimensions				
a, b, c (Å)	171.079, 51.033, 106.863	170.692, 50.976, 106.699	170.663, 50.9, 106.859	170.955, 50.933, 106.597
α, β, γ (°)	90.00, 112.589, 90.00	90.00, 113.232, 90.00	90.00, 112.535, 90.00	90.00, 113.446, 90.00
Resolution (Å)	51.71–2.37 (2.455–0.37)	48.48–1.67 (1.73–1.67)	49.35–1.92 (1.989–1.92)	48.9–1.96 (2.03–1.96)
*R_meas_*	0.2228 (1.029)	0.185 (1.579)	0.0815 (0.2692)	0.1696 (1.378)
*R_pim_*	0.08877 (0.4186)	0.1012 (0.9726)	0.0326 (0.1081)	0.0664 (0.5471)
Wavelength (Å)	1.000	1.000	1.000	1.000
Unique reflections	34970 (3456)	83673 (2177)	64491 (6470)	60844 (3489)
Completeness (%)	99.09 (96.59)	99.08 (92.07)	99.54 (98.11)	91.95 (57.41)
*<I>*/*σI*	4.49 (0.68)	6.27 (0.35)	11.04 (1.04)	5.29 (0.49)
CC1/2	0.988 (0.704)	0.975 (0.287)	0.998 (0.977)	0.996 (0.6)
Redundancy	6.2 (5.9)	3.0 (1.7)	6.0 (5.7)	6.4 (6.1)
Refinement				
*R_work_*/*R_free_*	0.2228/0.3239 (0.2598/0.3449)	0.2119/0.2699 (0.4972/0.6601)	0.1870/0.2508 (0.2541/0.3326)	0.2400/0.3162 (0.3901/0.4214)
Number of atoms				
Protein and ligand	6201	6439	6899	6199
Water	195	174	117	127
*B*-factors (Å^2^)				
All atoms	42.3	37.66	28.27	40.53
Solvent	42.15	40.53	35.8	38.88
R.m.s deviations				
Bonds (Å)	0.013	0.013	0.012	0.015
Angles (°)	1.37	1.28	1.24	1.41
Ramachandrans				
% Favored	92.88	95.93	95.93	91.86
% Outliers	0.76	0.76	1.15	0.42
Clashscore	18.32	8.78	9.98	15.76

**Table 2 ijms-23-05229-t002:** Steady-state kinetic parameters for WT-CTX-M-15 and S70A-CTX-M-15 enzymes with corresponding standard deviation (+/−).

Substrate	β-Lactamase	V_max_ (μmol/s)	K_m_ (μM)	k_cat_ (s^−1^)	k_cat_/K_m_ (μM^−1^ s^−1^)
Nitrocefin	WT-CTX-M-15	3.38 ± 0.12	1047.0 ± 75.8	338.3 ± 16.4	3.23 ± 0.02
	S70A-CTX-M-15	0.031 ± 0.003	45.6 ± 1.38	3.15 ± 0.17	0.07 ± 0.00
Ceftiofur	WT-CTX-M-15	3.14 ± 0.07	330.0 ± 14.1	313.7 ± 13.0	0.95 ± 0.01
	S70A-CTX-M-15	0.027 ± 0.00	35.0 ± 0.42	2.73 ± 0.08	0.08 ± 0.00
Ampicillin	WT-CTX-M-15	1.01 ± 0.06	197.8 ± 13.0	101.9 ± 5.6	0.52 ± 0.01
	S70A-CTX-M-15	0.007 ± 0.00	26.4 ± 3.7	0.67 ± 0.04	0.03 ± 0.00
DFC	WT-CTX-M-15	4.89 ± 0.2	141.2 ± 11.2	489.8 ± 16.2	3.46 ± 0.14
	S70A-CTX-M-15	0.006 ± 0.00	7.8 ± 0.03	0.59 ± 0.02	0.08 ± 0.00

**Table 3 ijms-23-05229-t003:** Thermodynamic parameters determined by isothermal titration calorimetry.

S70A-CTX-M-15	K_d_ (μM)	ΔH (cal mol^−1^)	ΔS (cal mol^−1^ K^−1^)	ΔG (kcal mol^−1^)
Clavulanic acid	288.6	−1333	11.7	−4.8
Nitrocefin	71.3	−2025	12.1	−5.7
Ceftiofur	38.2	−1157	16.3	−6.0
Ampicillin	28.8	−509	19.1	−6.2
DFC	10.7	−2246	15.2	−6.7

**Table 4 ijms-23-05229-t004:** Binding free energies calculated by molecular docking.

Substrate	ΔG _Binding CTX-M-15_ (kcal mol^−1^)	ΔG _Binding CTX-M-15 S70A_ (kcal mol^−1^)
Clavulanic acid	−6.2	−6.0
Nitrocefin	−6.7	−7.4
Ceftiofur	−6.8	−7.6
Ampicillin	−6.9	−7.4
DFC	−7.5	−8.6

## Data Availability

The structures discussed in this manuscript can be found at www.rcsb.org (accessed on 1 March 2022) deposited under the PDB ID: 7U48, 7U4B, 7U49, and 7U57.

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
