# Peer review of "Characterization of Interactions between CTX-M-15 and Clavulanic Acid, Desfuroylceftiofur, Ceftiofur, Ampicillin, and Nitrocefin"

_ijms, 2022, doi:10.3390/ijms23095229_

Round 1
Reviewer 1 Report
A comprehensive study of the mechanisms of interaction of beta-lactamases with antibiotics and inhibitors is of great interest for understanding the mechanisms of hydrolysis of antibiotics and the search for new inhibitors of these enzymes. Authors present a combined theoretical and computational study of the CTX-M-15 inhibition by several compounds. The major part of the work is related to the x-ray analysis. However, none of four x-ray structures ( PDB ID: 7U48, 7U4B, 7U49, and 7U57) is available for download. Therefore, we suggest to deposit the structures first and then resubmit the paper so that reviewers can consider the entire study.
Author Response
A comprehensive study of the mechanisms of interaction of beta-lactamases with antibiotics and inhibitors is of great interest for understanding the mechanisms of hydrolysis of antibiotics and the search for new inhibitors of these enzymes. Authors present a combined theoretical and computational study of the CTX-M-15 inhibition by several compounds. The major part of the work is related to the x-ray analysis. However, none of four x-ray structures ( PDB ID: 7U48, 7U4B, 7U49, and 7U57) is available for download. Therefore, we suggest to deposit the structures first and then resubmit the paper so that reviewers can consider the entire study.
It is the convention of the PDB for authors to hold the public release of the coordinates until the corresponding article is accepted for publication. Before accepting the coordinates and diffraction data, PDB performs a thorough validation check including geometry, chirality and electron density coverage. We have published more than 150 structural articles following this convention.
“Release Policies: PDB entries are processed by the members of the wwPDB (RCSB PDB, PDBe and PDBj). They are either released immediately (REL), when the corresponding paper is published (HPUB), or on a particular date (HOLD). https://deposit-2.wwpdb.org/deposition/pageview/depsummary”
Reviewer 2 Report
This is an interesting study in which the authors provide previously unexplored structural features of the interaction of CTX-M enzymes with antibiotics like ampicillin, ceftiofur and clavulanic acid, among others.
It is interesting that the authors were able to obtain acyl-complexes with clavulanic acid, which is important to assess its interaction with these important enzymes.
There are however a couple of main things I’d like to be further discussed in the text or clarified by the authors, and other minor issues.
Main comments:
- First, I wonder which was the rationale for selecting the antibiotics to be tested, and no others. For example, why were not tested ceftazidime and/or cefotaxime, considering these antibiotics are “signature” drugs to characterize the CTX-M ESBLs? Or why nitrocefin, provided that this antibiotic is not used clinically but only as reporter substrate?
- Second, if complex structure with clavulanic acid was obtained, inhibition kinetic parameters should be also informed to better evaluate the structural features of the interaction of CTX-M-15 with this inhibitor. Please, include the inhibition constants for clavulanic acid.
Other points to be addressed:
- Title: “CTX-M-15” must be written in uppercase letters.
- Line 67: Not all class B beta-lactamases are binuclear enzymes; in general, subclass B2 MBL are mono-zinc lactamases.
- Line 75: Class B lactamases have no activity at all against monobactams. In fact, they behave as inhibitors.
- Table 1: Rmerge is currently considered as a flawed indicator of data precision, and it is usually replaced by Rmeas or Rpim, which are more “realistic” values. I suggest replacing Rmerge by one of these other values.
- Lines 135-140: I wouldn’t expect succeeding in trapping acyl-complexes of CTX-M with nitrocefin and a third-generation cephalosporin (except for ceftazidime), due to the high turnover rates and high catalytic efficiency of these enzymes against those drugs. In fact, kinetic constants in Table 2 support this behavior, with kcat values of 300 1/sec and kcat/Km above 1 uM-1.sec-1.
- It is surprising that an ampicillin complex of CTX-M-15 was obtained! Considering that the kinetic parameters are so high for the wild-type variant (kcat = 102 1/sec and kcat/Km = 0.52 uM-1.sec-1), I would have expected the drug to be quickly hydrolyzed before freezing the crystals. Did the authors use a special protocol to avoid a rapid hydrolysis of ampicillin?
- Lines 250-251: what does it mean “higher catalytic efficiency” from the Michaelis-Menten plots? A higher Vmax? The term “catalytic efficiency” should be used carefully, as it implies not only the Vmax (directly related to the kcat) but also de Km … For example, the slope of the ascending curve in the MM plot is indicative of the kcat/Km (catalytic efficiency) only when the Km is much higher than the [S], so the term “[S]” can be omitted from the MM equation. Otherwise, I wouldn’t talk about “catalytic efficiency” from the plots.
- Table 2: “kcat” should be written with lower case “k”
- Line 284: a “-“ is missing in the deltaG value for ampicillin
- I wonder if the authors attempted to obtain crystallographic complexes of the Ser70Ala mutants with the different antibiotics …
- Figure 6 could be also included in the supplementary material
Author Response
Main comments:
- First, I wonder which was the rationale for selecting the antibiotics to be tested, and no others. For example, why were not tested ceftazidime and/or cefotaxime, considering these antibiotics are “signature” drugs to characterize the CTX-M ESBLs? Or why nitrocefin, provided that this antibiotic is not used clinically but only as reporter substrate?
These antibiotics were selected because ceftiofur and ampicillin are the most common antibiotics used in the Pacific Northwest dairy farms and ceftiofur, in particular, like many veterinary antibiotics have received little attention in the literature. Ceftiofur use has been attributed to selection for bacteria producing CTX-M and CMY-2 enzymes in these populations (Davis, M. A. et al. Recent Emergence of Escherichia coli with Cephalosporin Resistance Conferred by blaCTX-M on Washington State Dairy Farms. Appl Environ Microbiol 81, 4403-4410, doi:10.1128/AEM.00463-15 (2015), Liu, J. et al. Dairy farm soil presents distinct microbiota and varied prevalence of antibiotic resistance across housing areas. Environmental Pollution 254, 113058, doi:10.1016/j.envpol.2019.113058 (2019)). Ceftiofur is rapidly metabolized into DFC (~ 10 minutes after injection) (Hornish, R. E. & F., K. S. Cephalosporins in veterinary medicine - Ceftiofur use in food animals. Current Topics in Medicinal Chemistry, 717-731 (2002) and hence the interest in this structure. Nitrocefin was included to provide structural variation for our analysis of how the enzyme binds substrates. We added the Davis reference to the first sentence of the discussion (this was an accidental omission in the original submission).
- Second, if complex structure with clavulanic acid was obtained, inhibition kinetic parameters should be also informed to better evaluate the structural features of the interaction of CTX-M-15 with this inhibitor. Please, include the inhibition constants for clavulanic acid.
We inserted the Ki value of clavulanic acid in the revised manuscript. In addition, its double reciprocal plot has been added to the supplementary material (S5).
Other points to be addressed:
- Title: “CTX-M-15” must be written in uppercase letters.
corrected.
- Line 67: Not all class B beta-lactamases are binuclear enzymes; in general, subclass B2 MBL are mono-zinc lactamases.
The corresponding sentence has been rewritten as follow to accommodate reviewer’s concern.
“Class B β-lactamases are zinc metalloenzymes that contain one or two zinc ions in the catalytic site.”
- Line 75: Class B lactamases have no activity at all against monobactams. In fact, they behave as inhibitors.
We concur that monobactams are not efficient substrates for MBLs and this allows them to be employed as inhibitors. https://www.ncbi.nlm.nih.gov/pmc/articles/PMC4968164.
The corresponding previous sentence has been rewritten as following.
“Class B has notable activity against cephalosporins, carbapenems, and penicillins, although they are inhibited by monobactams [21].”
- Table 1: Rmerge is currently considered as a flawed indicator of data precision, and it is usually replaced by Rmeas or Rpim, which are more “realistic” values. I suggest replacing Rmerge by one of these other values.
Rmerge has been replaced by both Rmeas and Rpim. Thank you for calling attention to this detail.
- Lines 135-140: I wouldn’t expect succeeding in trapping acyl-complexes of CTX-M with nitrocefin and a third-generation cephalosporin (except for ceftazidime), due to the high turnover rates and high catalytic efficiency of these enzymes against those drugs. In fact, kinetic constants in Table 2 support this behavior, with kcat values of 300 1/sec and kcat/Km above 1 uM-1.sec-1.
The crystals were soaked in the artificial mother liquor (300 mM NH4SO4 and 30% PEG 3350) containing 30% glycerol and and flash-freezed. The pH of the solution was 5.5. This solution containing high concentration of NH4+ and SO42- ions in a liquid N2temperature is a severely suboptimal condition for enzyme turnover rate and this allowed the experiment to succeed.
- It is surprising that an ampicillin complex of CTX-M-15 was obtained! Considering that the kinetic parameters are so high for the wild-type variant (kcat = 102 1/sec and kcat/Km = 0.52 uM-1.sec-1), I would have expected the drug to be quickly hydrolyzed before freezing the crystals. Did the authors use a special protocol to avoid a rapid hydrolysis of ampicillin?
The reaction buffer for measuring kinetic parameter was the PBS (157 mM Na+, 140mM Cl−, 4.45mM K+, 10.1 mM HPO42−, 1.76 mM H2PO4−) that is pH 7.4. The crystals were soaked in the artificial mother liquor (300mM NH4SO4 and 30% PEG 3350) containing 30% glycerol and flash-freezed. The pH of the solution was 5.5. This solution containing high concentration of NH4+ and SO42- ions in a liquid N2 temperature is a severely suboptimal condition for enzyme turnover rate and this allowed the experiment to succeed.
- Lines 250-251: what does it mean “higher catalytic efficiency” from the Michaelis-Menten plots? A higher Vmax? The term “catalytic efficiency” should be used carefully, as it implies not only the Vmax (directly related to the kcat) but also de Km … For example, the slope of the ascending curve in the MM plot is indicative of the kcat/Km (catalytic efficiency) only when the Km is much higher than the [S], so the term “[S]” can be omitted from the MM equation. Otherwise, I wouldn’t talk about “catalytic efficiency” from the plots.
In revision, the corresponding statement was rewritten referring Table 2.
“The corresponding Vmax, Km, kcat for the four β-lactams are found in Table 2.”
- Table 2: “kcat” should be written with lower case “k”.
Corrected.
- Line 284: a “-“ is missing in the deltaG value for ampicillin.
Corrected.
- I wonder if the authors attempted to obtain crystallographic complexes of the Ser70Ala mutants with the different antibiotics …
Unfortunately, S70A was not crystalized despite our numerous attempts. We are still trying with no success.
- Figure 6 could be also included in the supplementary material:
Figure 6 has been moved to supplementary section, as suggested by reviewer.
Round 2
Reviewer 1 Report
It is difficult for me to agree with the authors' explanations about the refusal to deposit structures to the PDB. The procedure has long been debugged: before the publication of the paper, the structures are being kept in the status “Hold for release” and are available only to reviewers. For the reviewer, these data are very important and allow to analyze the conclusions. It is also possible to present some important data in the Supplementary materials.
Based on the material presented in the manuscript, I have a number of significant comments presented below:
- Figure 2(a): what is "! Domain" and "! /"Domain"
- Figure 3. The contour value of 1 sigma is too low. You should also show on the Figure electron density isosurfaces for contour values for 2 and 3 sigma. Is clavulanic acid visible at 2 and 3 sigma?
- Authors state that "Although association and breakage of clavulanic acid did not seem to produce any major changes on the overall structure of CTX-M-15, close inspection of the superimposed apo- and complex structure showed several subtle but significant structural adjustments." As no PDB structures are available, authors should demonstrate superimposition of the apo-form and CTX-M-15 - clavulanic acid complex
- The same questions as 2 and 3, but for complex with ampicillin.
- The same questions as 2 and 3, but for complex with DFC.
- From the description of the methods, it is not clear how the concentration of the enzyme’ active sites was determined (for Vmax) (Table 2). Determination by protein concentration in this case is incorrect.
- Some results of the kinetic analysis of the wild-type enzyme and its mutant (Table 2) require comments from the authors. The catalytic parameters for CTX-M-15 with respect to ampicillin and nitrocefin have been measured earlier. In this work, significantly worse Km values were obtained for the wild-type enzyme, while binding to the mutant form was improved markedly. This needs to be discussed and explained.
- The standard deviations of the catalytic parameters in the Table 2 are very small, while the kinetic data in Fig. S4 and C5 are characterized by a rather large deviation from the approximating curves. Need to be checked.
- It is well known for class A beta-lactamases, that S70 is a nucleophilic residue and it is responsible for the catalysis. In this manuscript, kcat for S70A mutant was decreased, but its value of 3 s-1 is relatively large indicating that the hydrolysis occurs. The author should comment on this issue about the probable mechanism of hydrolysis.
- It is also necessary to clarify the significantly smaller effects of reducing kcat/Km for the mutant (by 10-50 times) compared to those previously observed for class A beta-lactamases of other types (for example, by 9000 times for the TEM type enzyme; Stojanoski 2016).
- I would suggest eliminating speculations on the reaction and inhibition mechanisms (pp. 14-16 ), as generally it is already known for enzymes of this type and crystal structures obtained in this study do not provide any new details.
- lines 38-39 Please check the term extended-spectrum cephalosporin-resistant β-lactamases (ESCrE), should it be Extended-spectrum cephalosporin-resistant Enterobacteriaceae (ESCRE)?
- lines 71-73 The phrase is too general. It cannot be claimed that all class A beta-lactamases hydrolyze 3-4 generation cephalosporins and carbapenems as formulated. It is better to say that class A is quite heterogeneous in substrate specificity and to describe the substrate specificity of the CTX-M type enzymes that are the subject of study.

Author Response
Review 1: It is difficult for me to agree with the authors' explanations about the refusal to deposit structures to the PDB. The procedure has long been debugged: before the publication of the paper, the structures are being kept in the status “Hold for release” and are available only to reviewers. For the reviewer, these data are very important and allow to analyze the conclusions. It is also possible to present some important data in the Supplementary materials.
The three coordinates are uploaded for reviewer’s view.
Based on the material presented in the manuscript, I have a number of significant comments presented below:
1. Figure 2(a): what is "! Domain" and "! /"Domain"
This appears to be a web conversion error during submission and is corrected as following.
“α Domain and α/β Domain”.
2. Figure 3. The contour value of 1 sigma is too low. You should also show on the Figure electron density isosurfaces for contour values for 2 and 3 sigma. Is clavulanic acid visible at 2 and 3 sigma?
Contour level 3.0, 2.0, and 1.0 figures were made and inserted as supplementary materials, Figure S7.
Line 158-160: “The 2Fo-Fc map of covalently bound clavulanic acid at different counter levels 3.0, 2.0, and 1.0 sigma are shown in Figure S7.”
3. Authors state that "Although association and breakage of clavulanic acid did not seem to produce any major changes on the overall structure of CTX-M-15, close inspection of the superimposed apo- and complex structure showed several subtle but significant structural adjustments." As no PDB structures are available, authors should demonstrate superimposition of the apo-form and CTX-M-15 - clavulanic acid complex
The superimposition figure was made and inserted as supplementary materials, Figure S8.
Line 161-163: “close inspection of the superimposed apo- and complex structure showed several subtle but significant structural adjustments (Figure S8).”
4. The same questions as 2 and 3, but for complex with ampicillin.
Contour levels 3.0, 2.0, and 1.0 figures were made and inserted as supplementary materials Figure S9.
The superimposition figure was made and inserted as supplementary materials, Figure S10.
Line 191-192: “The 2Fo-Fc map of covalently bound ampicillin at different counter levels 3.0, 2.0, and 1.0 sigma are shown in Figure S9.”
5. The same questions as 2 and 3, but for complex with DFC.
Contour level 3.0, 2.0, and 1.0 figures were made and inserted as supplementary materials, Figure S11.
The superimposition figure was made and inserted as supplementary materials, Figure S12.
Line 223-225: “The electron density map of covalently bound DFC showed at contour level 1.0 sigma (Figure 5a) and contour level 3.0, 2.0, and 1.0 sigma at different orientation in the supplementary materials (Figure S11).”
6. From the description of the methods, it is not clear how the concentration of the enzyme’ active sites was determined (for Vmax) (Table 2). Determination by protein concentration in this case is incorrect.
To be clear, we have rewritten the corresponding method section as following.
Line 616- 619: “The reaction was initiated by adding 75 ml of stock solutions of wildtype and mutant enzymes (40 nM) to 225 ml reaction buffer resulting in a final enzyme concentration of 10 nM. The enzyme concentration of the stock solution was measured by Bradford assay.”
7. Some results of the kinetic analysis of the wild-type enzyme and its mutant (Table 2) require comments from the authors. The catalytic parameters for CTX-M-15 with respect to ampicillin and nitrocefin have been measured earlier. In this work, significantly worse Km values were obtained for the wild-type enzyme, while binding to the mutant form was improved markedly. This needs to be discussed and explained.
The kinetic experiments in the “Faheem, M., et al 2013” used 50 mM Na phosphate buffer, pH 7.0 at 30oC. In our experiment, PBS (157 mM Na+, 140mM Cl−, 4.45mM K+, 10.1 mM HPO42−, 1.76 mM H2PO4−, pH 7.4) was used at 37oC system. Thus, there are differences in buffered system, pH and temperature. In addition, Faheem et al. used His-tagged with one-step purification with Ni-NTA. We had to use 2-step purification (anion-exchange column Hiprep DEAE Sepharose 16/10 (GE Healthcare) and hydroxyapatite column (Bio-Rad )) due to being a non-tagged wild-type enzyme. Apparently, two enzymes, despite same name, show only 41% identity explaining a part of differences in their kinetic parameters. GenBank accession number AY044436.1 for CTX-M-15 in our study and GenBank accession number JN860195.1 for Faheem’s enzyme. We have inserted the GenBank accession number to make it clear.
Line 110-112; “The apo- and complex crystal structures of wild-type CTX-M-15 (GenBank accession number AY044436.1) were obtained at high resolution allowing in-depth studies of conformational changes upon substrate binding (Table 1).”
8. The standard deviations of the catalytic parameters in the Table 2 are very small, while the kinetic data in Fig. S4 and C5 are characterized by a rather large deviation from the approximating curves. Need to be checked.
Standard deviations (SDs) in Table 2 and Fig S4 were verified again. For Table 2, the SD value is the average of all the SDs for all the 20 data points collected. In Fig S4, the SD represents each individual data point.
Line 267-268: “Table 2. Steady-state kinetic parameters for WT-CTX-M-15 and S70A-CTX-M-15 enzymes with corresponding standard deviation (+/-)”
9. It is well known for class A beta-lactamases, that S70 is a nucleophilic residue and it is responsible for the catalysis. In this manuscript, kcat for S70A mutant was decreased, but its value of 3 s-1 is relatively large indicating that the hydrolysis occurs. The author should comment on this issue about the probable mechanism of hydrolysis.
We have inserted the following sentence in response to the reviewer’s suggestion.
Line 391- 400: “It is apparent that hydrolysis of the mutants are minuscule (Figure 6). For all substrates tested, the turnover numbers are two-three orders of magnitude lower than the wildtype. Some minor hydrolytic activity still existed probably due to other S70-independent mechanism (covalent catalysis). One of the plausible mechanisms is that a nucleophilic attack may still proceed with a water molecule to produce a tetrahedral intermediate followed by a collapse to yield the hydrolyzed product as reported before for the serine protease. (Cater & Wells, Nature 332: 564–568). In addition, preferential binding of the transition state such as using oxyanionic hole could increase a local concentration of transition state as shown in general catalytic antibodies (Mader & Barlett, Chem. Rev. 97, 1281-1301).”
10. It is also necessary to clarify the significantly smaller effects of reducing kcat/Km for the mutant (by 10-50 times) compared to those previously observed for class A beta-lactamases of other types (for example, by 9000 times for the TEM type enzyme; Stojanoski 2016).
In the paper (Stojanoski, V. et al, Biochemistry. 55(2016) 2479–2490, the authors worked with six enzymes (both class A and D b-lactamases) and mentioned that the differences between the serine mutant and wild-type for most of those enzymes were 40 - 65-fold reduction for ampicillin, which are similar range as our case. TEM was uniquely ~9000-fold reduction for ampicillin, although it showed only a 92-fold reduction for cephalothin. CTX-M-15 and TEM are quite different enzymes with only 40% aa sequence identity and have different capacities for hydrolyzing different substrates. In addition, their assays were performed at 30°C in 50 mM sodium phosphate buffer pH 7.2.
11. I would suggest eliminating speculations on the reaction and inhibition mechanisms (pp. 14-16 ), as generally it is already known for enzymes of this type and crystal structures obtained in this study do not provide any new details.
The corresponding section for hydrolysis reaction has been significantly shortened and figure has been moved to the supplemental section. Considering the authentic finding of clavulanic acid through crystal structure, authors wish to keep it.
12. lines 38-39 Please check the term extended-spectrum cephalosporin-resistant β-lactamases (ESCrE), should it be Extended-spectrum cephalosporin-resistant Enterobacteriaceae (ESCRE)?
This has been corrected as following.
Line 37- 39: “CTX-M together with CMY are now the most prevalent Extended-spectrum cephalosporin-resistant Enterobacteriaceae (ESCRE) found in E. coli, K. pneumonia, and Proteus mirabilis [6].”
13. lines 71-73 The phrase is too general. It cannot be claimed that all class A beta-lactamases hydrolyze 3-4 generation cephalosporins and carbapenems as formulated. It is better to say that class A is quite heterogeneous in substrate specificity and to describe the substrate specificity of the CTX-M type enzymes that are the subject of study.
We did not intend to include all 3rd and 4th generation cephalosporins but indicated the substrates that CTX-M is able to hydrolyze such as monobactam and carbapenems (like 3rd generation), and cefepime (4th generation). We agree that CTX-M-15 cannot hydrolyze all 3rd and 4th generation cephalosporins but that statement was not claimed in this sentence in the paper – “Class A β-lactamases, including CTX-M, display a wide range of resistance to antibiotics such as penicillins, early cephalosporins, monobactams, carbenicillin, cefepime, and carbapenems.”